# Is There Any Relationship between Physical Activity Levels and Academic Achievement? A Cross-Cultural Study among Spanish and Chilean Adolescents

**DOI:** 10.3390/bs13030238

**Published:** 2023-03-08

**Authors:** Laura O. Gallardo, Diego Esteban-Torres, Sheila Rodríguez-Muñoz, Alberto Moreno-Doña, Alberto Abarca-Sos

**Affiliations:** 1Faculty of Social Sciences and Humanities, University of Zaragoza, C/Atarazana 4, 44003 Teruel, Spain; 2Faculty of Medicine, Universidad de Valparaíso, Casa Central—Angamos, Viña del Mar 655, Chile

**Keywords:** cross-cultural, physical activity, academic achievement, adolescents, socio-economic status, personal variables

## Abstract

The current scientific literature has shown significant disparity in results when physical activity is linked to academic achievement among adolescents. Thus, the main objectives of this study were (1) to analyze the relationship among students’ academic achievement, intention to be physically active, and physical activity levels depending on the country (Spain or Chile), as well as to analyze these relationships based on students’ socio-economic status, type of school, school year, gender, and body mass index; and (2) to analyze the differences between all these variables depending on the students’ country and gender. In total, 3052 adolescents participated in the research (14.58 ± 1.39 years): 734 Chilean students (336 males and 398 females) and 2318 Spanish students (1180 males and 1138 females). Various questionnaires were used to measure the study variables. The results revealed significant relationships between academic achievement and the rest of the variables in Spanish adolescents, but in the Chilean population, academic achievement was significantly related only to socio-economic status and the type of school. Moreover, Spanish students obtained higher scores, especially the males, except for academic achievement, which was higher in females. There were also significant differences in academic achievement, intention to be physically active, physical activity levels, and socio-economic status depending on the country, with all scores being higher in Spain. Given the results, the country seems to be an important factor when comparing academic achievement and physical activity levels, besides other demographic variables.

## 1. Introduction

Physical activity (PA) generates many health benefits [1]. Higher levels of PA are associated with increased cardiorespiratory endurance, better vascular and musculoskeletal function, and decreased fatigue [2]. These benefits are not only at a physical level but also at other levels, such as the psychological level. For example, a study carried out with European adolescents between 14 and 16 years of age confirmed that moderate PA contributes to greater general well-being, with lower levels of anxiety and depression in the participants [3]. Another psychological factor linked to PA is young people’s intention to be physically active in the future [4]. It has been shown that adolescents with a positive intention to be physically active score higher on physical tests such as endurance or speed [5].

At the cognitive level, some authors support the association between moderate–vigorous PA and an improvement in cognition [6]. These authors assess cognition using the results from neuropsychological exams, which consist of tests that address processing speed, memory, or executive function.

On the other hand, also linked to cognitive issues, the relationship between PA and academic achievement (AA) or academic results has also been researched. The studies carried out in recent years focusing on the adolescent population have shown disparate results.

In a review conducted by Herting and Chu, most studies agreed that adolescents´ daily PA was associated with their AA, cognitive function, brain structure, and activity [7]. Specifically, 48% of the selected studies showed significant effects of PA on cognition, and 60% showed significant effects of PA on AA [8]. Despite this, the percentage of studies that showed no significant effects was high. In another review developed by Singh et al., 80% of the articles showed a strong association between the variables, although the authors pointed out that the relationship with physical condition has been studied more than the relationship with general PA [9].

In the Spanish adolescent population, results showed a significant negative relationship between PA levels during the week and AA, and no relationship with PA on the weekend. However, the male group that reached levels of PA close to compliance with the recommendations developed by the US Department of Health and Human Services (60 min of daily moderate–vigorous PA) [10] was more likely to obtain high AA [11].

In Chilean adolescents, some investigations analyzed the relationship between general PA and AA, including grades obtained exclusively in the areas of mathematics and language. The results showed that adolescents with a medium–low level of PA, which coincided with the participants who presented with obesity, were less likely to achieve high grades [12].

Other studies with the Chilean adolescent population analyzed the relationship between PA levels and eating habits with general AA. The results indicated that healthy habits, PA, and good nutrition were associated with higher AA [13]. Young people who practice PA regularly are significantly less likely to be overweight or obese in adulthood [14].

Regarding these results, it is appropriate to carry out comparative studies between countries to analyze whether the relationship between PA and AA is different according to the country.

A study carried out in 42 countries confirmed a positive relationship between self-reported AA and PA, specifically in students who practiced between 5 and 6 days of moderate–vigorous PA [15]. The data had an inverted “U” shape, with lower AA in individuals who practiced either little or considerable PA. However, differences between various countries were not analyzed, and few studies seem to have addressed this issue. Nevertheless, another study found no significant differences between motor skills, executive function, and early achievement in children from six countries [16].

Some authors consider that there are several reasons for the variability of the results in the relationship between PA and AA. On the one hand, AA is usually measured through the academic results obtained, which may differ because the educational system may vary as a function of the country [17]. On the other hand, the lack of measures to compare these variables and the differences in the frequency of PA in different countries is noteworthy. Finally, the lack of control over other factors that could influence the results, such as family socio-economic status (SES), age, or psychological factors may condition the relationship between AA and PA [18]. One investigation found that higher gender inequality was associated with gender differences in PA. Likewise, lower levels of gender inequality were associated with increased female and male PA [19].

Consequently, although many studies have analyzed the relationship between PA and AA, few researchers have studied country differences in adolescence. In addition, the are two studies that were found who presented dissimilar results: in one of them, they showed significant relationships between PA and AA, and in the other, they did not [15,16].

Therefore, given the previous research, our main objectives of this study were 1) to analyze the relationship between the variables AA, intention to be physically active, and levels of PA depending on the country (Spain or Chile), taking into account other variables such as SES, type of educational center, grade, gender, and body mass index (BMI); and 2) to analyze possible differences by country in PA levels, AA, intention to be physically active, PA family SES, gender, and BMI. Based on these objectives, the following study hypotheses are established. (a) The relationship between all the variables will be significant and positive in both countries, except for BMI, which will be significant and negative. (b) The means will be higher and significant in Spain, except for BMI, which will be higher in Chile. In addition, the means will be higher in males except for AA, which will be higher in females. There will be significant differences between the two countries as a function of gender in all the study variables.

## 2. Materials and Methods

### 2.1. Participants

The sample comprised 3052 adolescents: 734 from Chile, Valparaiso region, (*M* = 14.74, *SD* = 1.47 years) and 2318 from Spain, Autonomous Community of Aragon (*M* = 14.53, *SD* = 1.37 years). From Chile, there were 336 boys, aged 14.80 ± 1.47 years (BMI = 20.97 ± 3.45 kg/m^2^), and 398 girls, aged 14.70 ± 1.47 years (BMI = 21.75 ± 3.85 kg/m^2^), from seventh and eighth basic grades and from first to fourth half grades from six high schools: three concerted and three public. From Spain, there were 1180 boys, aged 14.53 ± 1.38 years (BMI = 20.40 ± 3.56 kg/m^2^), and 1138 girls, aged 14.52 ± 1.36 years (BMI = 20.02 ± 2.86 kg/m^2^), in the first to the fifth grades of secondary education from fourteen high schools: three concerted and eleven public. The samples of schools, from Chile and Spain, were chosen by convenience based on accessibility and willingness to cooperate. Data were collected during the academic course of the year 2016.

### 2.2. Measures

**Demographic Variables.** The questionnaire included a personal data section: gender; grade (ranging from first year of secondary school to second year of high school; a total of 6 courses); type of school (1 = public or municipal, 2 = concerted or privately subsidized, and 3 = private); students’ weight; and height.

**Academic Achievement.** AA was calculated using the arithmetic mean obtained in the teaching areas of the previous year. The scores were reported by the participants, a method used in other studies [20]. In this case, AA was calculated with a question asking the average on a 10-point scale.

**Physical Activity Levels**. The “International Physical Activity Questionnaire-Short Form” (IPAQ-SF) was used to measure PA [21]. It was validated in the adolescent population and in Spanish adolescents [22]. It provides information on the PA carried out by each individual over the last seven days and also asks about the intensity (light, moderate, or vigorous), the frequency (days per week), and the duration (time per day). It presents seven items (e.g., During the last seven days, on how many days did you perform vigorous physical activities such as lifting heavy objects, playing sports intensively, running or cycling fast?). The responses indicate the days of practice (from zero to seven) and the time invested (in minutes). The IPAQ-SF showed negligible to small correlations with total activity level measured with objective devices (median = 0.29).

**Intention To Be Physically Active**. The questionnaire “Intention to be Physically Active Measurement” was validated in the Spanish population [23,24]. It assesses post-high school intention to be physically active. It consists of five items (e.g., “After finishing high school, I would like to be part (or continue) of a sports training club”), which are rated on a five-point Likert-type scale, ranging from 1 (totally disagree) to 5 (totally agree). The final score is obtained by calculating the arithmetic mean of the scores obtained in each item, so the higher the score, the greater the intention to be physically active in the future. The construct validation data provided a reliability (Cronbach´s alpha coefficient) of 0.94. In both our samples, the reliability was *α* = 0.813 for the Spanish sample and *α* = 0.813 for the Chilean one. 

**Family Socio-economic Status**. The “Family Affluence Scale II” (FASII) was used to compare family SES between countries [25]. It consists of 4 items, with response variation and, consequently, score variation (e.g., “Does your family have a car, van or truck?” No [zero]; Yes, one [one]; Yes. Two or more [two]). The final score, called the FAS index, is the sum of the scores of the answers to each question on a scale of zero to nine points. The higher the score, the higher the family’s SES level. The internal consistency in our study recorded a moderate association (α = 0.61). However, this result is similar to previous studies [26].

### 2.3. Procedure

The procedure in the present investigation was the same for both countries.

Firstly, we note that this study was approved by CEICA (Ethics Committee of the Autonomous Community of Aragon).

Then, high-school managers were contacted to request the schools´ participation. Once the participation was confirmed, a circular was provided to the teachers, families, and participants to inform them about the protocol and aims of the study. It included consent for participation by the parents/guardians and assent for the students themselves. Researchers made an appointment with the high schools to administer the questionnaires. Questionnaire completion was carried out during regular class time under the supervision of at least a researcher and the teacher in the classroom. The data were coded assigning a personal identification to each participant.

The need for a sincere response was highlighted, with an explanation that the data reported were used exclusively for research purposes.

### 2.4. Data Analysis

To confirm sample distribution normality, the Kolmogórov–Smirnov test was performed. Once normality was confirmed, a Pearson bivariate correlation analysis was carried out with all the variables (PA levels, AA, intention to be physically active, SES, BMI, gender, type of school, and school year) to determine their relationships independently in each country. Subsequently, multivariate analysis (MANOVA) was performed according to the country with the entire sample and with both sexes separately, in addition to descriptive statistics to determine the means and standard deviation.

All analyses were performed using the statistical software SPSS version 26.0.

## 3. Results

First, the results of the bivariate correlation analyses performed between all the study variables are presented. Table 1 shows the relationships between variables in the Spanish population. There were significant correlations between most variables: AA was related to intention to be physically active, SES, and school. PA was related to intention to be physically active, SES, center, and gender. Intention to be physically active correlated with SES and gender. Grade was related to school and BMI. SES correlated with school, and BMI correlated with gender. The highest values were between the PA levels and the intention to be physically active (*r* = 0.41, *p* < 0.01) and between BMI and grade (*r* = 0.29, *p* < 0.01). 

Table 2 presents the results of the correlations between the previously presented variables for the Chilean population. The highest correlations were again found between PA and the intention to be physically active (*r* = 0.42, *p* < 0.01), followed by PA and gender (*r* = 0.22, *p* < 0.01) and the type of school with AA and SES (*r* = 0.306, *p* < 0.01 and *r* = 0.39, *p* < 0.01, respectively).

The results in Table 3, firstly considering the total sample, reveal the significant differences in AA, intention to be physically active, PA levels, and SES depending on the country, with Spain obtaining the highest values. Regarding BMI, significant differences were found, although Chile presented higher values. This means that, in our study, the country has a statistically significant effect on all variables, as shown by *p* < 0.001.

Secondly, considering the female gender, differences were found in almost all the target variables. All the Spanish female adolescents scored higher in AA, intention to be physically active, PA levels, and SES. However, Chilean female adolescents scored higher in BMI. Concerning male gender, significant differences were found in AA, intention to be physically active, and SES. Again, Spanish male adolescents scored higher. Nevertheless, our results revealed no significant differences in PA and BMI in the male gender.

Multivariate analysis was used to identify the differences in the studied variables by country and gender. Our results showed a significant effect of the country for the entire population studied (Wilks’ λ = 0.847, F(1, 1847) = 66.533, *p* < 0.001, η^2^ = 0.020), for males (Wilks λ = 0.883, F(1, 918) = 24.136, *p* < 0.001, η^2^ = 0.117), and for females (Wilks λ = 0.793, F(1, 927) = 48.128, *p* < 0.001, η^2^ = 0.207).

## 4. Discussion

The first hypothesis proposed in this research stated that: “the relationship between all the variables would be significant and positive in both countries, except for BMI, which would be significant and negative.” The results of Spain showed a significant relationship between most of the variables, whereas in Chile, hardly any relationships were observed.

The AA was significantly related to all the study variables in the Spanish population, but it was related only to family SES and the type of school in the Chilean population.

Data on the relationship between AA and PA levels are inconclusive. These results are supported by other studies, such as those included in the review reporting that 60% of all the articles that met the established selection criteria found a positive relationship between the two variables [8]. Based on the review carried out on young people aged between 6 and 18 years, 10 of the 16 selected articles evaluating the relationship between self-reported PA and AA showed positive associations [17]. According to these authors, this variation in results may be due to the different measurement methods used in the investigations, both in PA and AA.

On the other hand, we highlight the SES and the type of school, which are significantly and positively related to AA in both countries. Although our samples are not representative, these results are reinforced by some studies finding that parents’ higher SES is related to their children’s AA [27]. However, some authors point out that this relationship between AA and SES could differ in different social, economic, and cultural contexts [28].

The PA levels in the Spanish sample are also related to all the variables except for BMI and in the Chilean sample, with the exception of grade and BMI. The strongest correlations were between PA and the intention to be physically active, both in the Spanish and the Chilean samples. This may be because the intention to be physically active is a strong predictor of PA levels, and there are studies in which intention has explained up to 43% of the variance of PA [29]. Similar results have been found in other studies, revealing that variables such as gender and age are related to PA and are significant predictors of the degree of compliance with the recommendations [30]. In this case, the relationship between PA levels and age was significantly negative in the Spanish population, which can be explained by the fact that across adolescence, other interests arise, generating new life habits, including a decrease in PA [30]. On the other hand, although some studies included BMI as a predictor, others did not show a relationship between BMI and PA [30,31,32,33]. Therefore, we can conclude that there is no unanimity in the data reported to date about a possible relationship between PA and BMI.

The type of school (public, private, or concerted) in which adolescents study and the family´s SES are also related to each other and to the PA levels of the Spanish and Chilean adolescents. This could be due to the fact that, generally, young people with a high family SES reside in environments that promote the practice of PA [34]. A review carried out on the adolescent population found that young people with a high SES were less likely to be sedentary [35]. In countries with a medium–low SES, the opposite occurs: young people with a high SES tend to be more sedentary. These data do not agree with the present investigation in the Chilean population.

It should be noted that young people with a high family SES tend to attend private fee-paying schools [36]. Private schools usually have smaller class sizes and better teaching resources and facilities, which can create a different classroom culture [37]. The literature indicates that the relationship between the type of school and PA is poorly understood. However, it also reports that schools, both public and private, provide a broad variety of opportunities for PA [38]. Nonetheless, there is evidence that the current school efforts to increase PA levels do not impact young people’s daily PA positively, regardless of gender or SES [39].

The significant relationship in both countries between the intention to be physically active and the demographic variables of gender, age, and SES (except for age in the Chilean population) shows that all these variables influence the greater or lesser intention to practice [5].

The second hypothesis stated that “the means would be higher and significant in Spain, except for BMI, which would be higher in Chile. In addition, they will be higher in males except for AA, which would be higher in females. There would be significant gender differences between the two countries in all the study variables.” The results confirmed this hypothesis, except for BMI, which did not present significant differences. In the rest of the variables, there were differences between the two countries, with the averages being higher in Spain and in males, except AA, which was higher in females.

The results presented are supported by other studies that also confirm that AA is higher in females [40]. This could be explained because girls have higher school productivity, taking much more advantage of study hours than boys [41]. Likewise, it has been found that girls have a greater intention to continue studying for more years, a factor that could also significantly account for these data.

Differences in AA have also been found depending on the country. Some studies have analyzed family income disparity in AA, concluding that there are differences, with higher AA in more prosperous communities [42]. It should be noted that AA is considered an important predictor of the quality and equity of education when comparing countries [43].

It was also verified that PA levels are higher in males in both countries. The gender difference in PA has also been reported in other studies [44]. In fact, this gender disparity is seen in many countries, perhaps because most countries still maintain traditional gender roles [45,46]. Similarly, these results show that the social and cultural context are important influencing agents in female adolescents’ PA experiences [47]. In contrast, Chilean boys’ PA levels were also lower, consistent with results obtained in previous studies with this population [48].

Regarding the intention to be physically active, there were differences between the two countries, with higher averages in males. Other studies have evaluated this variable according to gender, also reporting higher levels in males [49]. In addition, intention has been compared in young Spanish and Latin Americans, but the results are mixed. Some authors indicate that Spanish adolescents express a higher intention to be physically active than Argentines, whereas other studies report that Colombian and Ecuadorian adolescents have a higher rate of intentionality than Spaniards [47].

SES also presented differences between countries, although it should be noted that these differences were not observed when the samples of participants were analyzed separately by gender. The means were still higher in the Spanish population, which could be explained by the level of the development of the two countries, as some authors consider Chile a developing country [48].

The BMI indices of the studied population are between percentiles 18.5 and 24.9, which are considered normal. These data are positive, as there is currently some obesity in the adolescent group, reaching 38% overweight in Chile and 26% in Spain [50,51]. In our study, BMI was higher in the Chilean population in both genders than in Spanish adolescents, in line with the data from previous investigations [52].

## 5. Conclusions, Limitations, and Prospects

### 5.1. Conclusions

In conclusion, regarding the objectives, we can affirm that both AA and PA levels are related to each other and to other factors such as the intention to be physically active in the future, gender, age, and SES. BMI is not related to PA levels in both countries. The strongest correlations were between PA and the intention to be physically active, both in the Spanish sample and the Chilean sample, because it is a strong predictor of PA.

The country should be highlighted as an important factor, because the relationships between the variables differ depending on the population studied. In Spain, there were significant relationships between most of the variables, whereas in Chile, SES and the type of school were significantly and positively related to AA but no other variables. Likewise, significant differences are shown in AA, PA, the intention to be physically active, and family SES when participants from both countries (Chile and Spain) were compared, with higher means in the Spanish population. These differences are present in the total population, in both the male and female populations, so we can state that culture is a key factor when analyzing variables such as PA or AA.

### 5.2. Limitations

First, we point out that all the study variables were measured subjectively, as they were reported by the participants. Therefore, some studies have shown that the results of the relationships between variables such as AA and PA differ depending on whether they were evaluated objectively or subjectively [17]. This may be because subjective data are not as accurate as objective data, such as PA levels measured by accelerometry [52]. Likewise, subjective data on PA tend to overestimate reality [50]. Height and weight values were also self-reported by the participants. Although these data have been found to correlate highly with objective values, adolescents may tend to report higher height and lower weight than the actual data [51].

On the other hand, this is not a longitudinal study, which would strengthen the results obtained by comparing them at different time points. In addition, the data collection of the total sample was not carried out during the same period, so there may be a few months’ difference between measurements. This could have conditioned the PA levels of the population studied.

Finally, despite the fact that the sample participating in this study is quite large, it is not a representative sample of each region. If this were the case, the results could be generalized to the entire studied population. Moreover, the differences from rural or urban environments have not been registered, showing a limitation of the present study.

### 5.3. Prospects

These results can be very useful in designing and implementing a multifactorial intervention program based on improving AA. A specific program could be designed for each country based on these results, thus adapting to the cultural demands of each population. This would also contribute to improving PA levels in adolescents because youth is the decisive moment for improving future healthy behaviors [53].

## Figures and Tables

**Table 1 behavsci-13-00238-t001:** Correlations between the study variables in the Spanish population.

	AA	PA	Intention	Grade	SES	School	BMI	Gender
AA	-	0.054 *	0.137 **	−0.068 **	0.161 **	0.079 **	−0.159 **	−0.084 **
PA		-	0.411 **	−0.110 **	0.093 **	0.070 **	−0.016	0.137 **
Intention			-	−0.091 **	0.113 **	0.037	−0.087 **	0.057 **
Grade				-	−0.032	0.107 **	0.285 **	−0.026
SES					-	0.069 **	−0.156 **	−0.016
School						-	0.010	0.023
BMI							-	0.058 **
Gender								-

AA = academic achievement; PA = physical activity; SES = socio-economic status; BMI = body mass index. ** Significant correlation at the 0.01 level (bilateral). * Significant correlation at the 0.05 level (bilateral).

**Table 2 behavsci-13-00238-t002:** Correlations between the study variables with the Chilean population.

	AA	PA	Intention	Grade	SES	School	BMI	Gender
AA	-	−0.015	−0.018	−0.002	0.178 **	0.306 **	−0.082	−0.048
PA		-	0.422 **	−0.043	0.187 **	0.140 **	−0.090	0.224 **
Intention			-	−0.041	0.167 **	0.135 **	−0.073	0.113 **
School				-	−0.022	−0.037	0.001	0.000
SES					-	0.389 **	−0.044	0.082
Center						-	−0.046	0.058
BMI							-	0.43
Gender								-

AA = academic achievement; PA = physical activity; SES = socio-economic status; BMI = body mass index. ** Significant correlation at the 0.01 level (bilateral).

**Table 3 behavsci-13-00238-t003:** Descriptives and MANOVA of physical activity levels and academic achievement according to country and gender.

Sample	Variable	Spain	Chile	*F*	*p*	η^2^
Total sample	Academic achievement	68.30 (14.10) a	56.99 (7.41) a	247.149	<0.001	0.118
	Intention	4.08 (0.85) a	3.78 (0.95) a	38.321	<0.001	0.020
	PA	52.20 (52.55) a	41.14 (44.38) a	15.233	<0.001	0.008
	SES	6.43 (1.63) a	5.39 (2.07) a	116.243	<0.001	0.059
	BMI	20.29 (3.21)	21.28 (3.55)	29.072	<0.001	0.016
Male gender	Academic achievement	66.92 (13.97) a	56.96 (7.54) a	91.842	<0.001	0.091
	Intention	4.14 (0.83) a	3.88 (0.94) a	14.840	<0.001	0.016
	PA	58.60 (53.66) c	50.50 (53.40) c	3.509	>0.05	0.004
	SES	6.38 (1.68) a	5.52 (2.09) a	35.940	<0.001	0.038
	BMI	20.57 (3.53)	20.92 (3.44)	1.569	>0.05	0.002
Female gender	Academic achievement	69.73 (14.10) a	57.01 (7.31) a	165.287	<0.001	0.151
	Intention	4.02 (0.87) a	3.69 (0.96) a	22.322	<0.001	0.024
	PA	45.66 (50.63) c	32.85 (32.42) c	12.486	<0.001	0.013
	SES	6.49 (1.58) a	5.27 (2.04) a	85.995	<0.001	0.085
	BMI	20.01(2.82)	21.60(3.63)	46.024	<0.001	0.047

PA = physical activity; SES = socio-economic status; BMI = body mass index. There are significant differences between countries based on gender: a (*p* < 0.001), c (*p* < 0.05).

## Data Availability

The data presented in this study are available on request from the corresponding author. The data are not publicly available due participants are minors.

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
