# Peer review of "Is There Any Relationship between Physical Activity Levels and Academic Achievement? A Cross-Cultural Study among Spanish and Chilean Adolescents"

_behavsci, 2023, doi:10.3390/bs13030238_

Round 1

Reviewer 1 Report

I have made comments below which need to be addressed to improve the quality of your manuscript.

Abstract: The second sentence is too long. Please break it up into two sentences and write more concisely.

Introduction

Page 1: “…adolescents between 14 and 16 years confirms that moderate PA contributes to greater general

Page 2: “In a review, most studies…..” – Please include the lead author's name when citing this review.

Page 2: “In another review, 80%...” Again, please include the lead author's name when citing this review.

Page 2: “…recommendations (60 minutes of daily moderate-vigorous PA….” – please cite the exercise guidelines being stated here.

Page 2: “We highlight that childhood obesity usually persists in the future….” – please revise because this sentence makes no sense.

Page 2: “…associated with increased female and male PA.”

Page 2: “Therefore, the main objectives of this study was to 1)…..”

Page 3: This should be a new sentence: “2) To analyze possible differences by country in PA levels, AA, intention to be physically active, PA family SES, gender, and BMI.

Materials and Methods

No idea what these numbers are referring to “M = 14.74, SD = 1.47”. Please add units.

Use past tense when describing what was done and the participants included. E.g., “From Spain, there were 1180 boys…”

Procedure

“….appointment with the high schools to administer the questionnaires.”

Discussion

“…self-reported PA and AR showed positive associations…” – What is “AR”?

Page 8, line 133: “….consider Chile a developing country [47].”

Page 8, line 140: “…aged between 12 and 15 years…”

Page 8, line 143: “…Chilean population in both genders that Spanish adolescents….” – Very poor English grammar. Please revise.

Author Response

We would like to thank the reviewers for the quality of their comments, which have been extremely helpful in producing a strengthened version of our manuscript. We also give thanks to the editor for the invitation to deal with all the reviewers’ comments and resubmit an updated version of our manuscript. Please see the attachment: we provide point-by-point responses to the comments carried out by reviewer 1. In this response, we indicate the modifications carried out in the revised version of the manuscript. The changes have been highlighted in green in order to facilitate the process of review. We hope the manuscript is now acceptable for publication in Behavioral Sciences.

Reviewer 2 Report

The paper deals with a theme relevant to the journal.

The sample size and the instruments are adequate to achieve the proposed objectives.

The theoretical introduction and discussion are correctly adjusted to the proposed objectives and results obtained.

The results are clearly stated.

Below I propose some changes that I consider necessary before the publication of the paper.

1. It must be specified how the participating schools were selected.

If there are biases in this selection, the results would be biased.

This could cast doubt on the validity of the results.

2. Was it checked whether the participants live in a rural or urban environment?

This variable is relevant in terms of lifestyle and the possibilities of physical activity.

If it was controlled, it should be specified.

If it was not controlled, it should be indicated as a limitation.

3. The data may be representative of the regions of Aragón and Valparaíso.

But they are not representative of Spain and Chile.

This should be taken into account when interpreting the results.

Important changes should be made to the discussion in this regard.

4. Hypothesis.

The hypotheses must be justified.

If hypotheses are not justified, it would be preferable to delete them.

5. Formal aspects.

5.1. When the possible answers to the demographic questionnaire are indicated, it is not necessary to write the number, only the content of the options.

That is, it is not necessary to write "1=first year of secondary education, 2=second year of secondary education,..."

It would suffice to write "ranging from first year of secondary education to second year of high school".

5.2. In values ​​ranging from 0-1, such as p-values, the "0" is not indicated.

Only the decimal part of the number is indicated. 

That is, p= 0.024 is incorrect.

Instead, p= .024 is correct.

Author Response

We would like to thank the reviewers for the quality of their comments, which have been extremely helpful in producing a strengthened version of our manuscript. We also give thanks to the editor for the invitation to deal with all the reviewers’ comments and resubmit an updated version of our manuscript. Please see the attachment:  we provide point-by-point responses to the comments carried out by the reviewer 2. In this response, we indicate the modifications carried out in the revised version of the manuscript. The changes have been highlighted in green in order to facilitate the process of review. We hope the manuscript is now acceptable for publication in Behavioral Sciences.

Reviewer 3 Report

Introduction -

What are the common elements for which Spanish and Chilean adolescents were selected for the study? Mention the criteria that were the basis of this comparative study from the perspective of the two categories of teenagers?

We recommend that you emphasize the novelty of the study in correlation with previous studies.

Materials and Methods

The Study design section is missing, where we recommend you highlight the period of the study and the way the study was conducted.

The very large difference between the numbers of the two samples can determine the unsustainability of the results and conclusions.

Results

Statistical processing is insufficient. Questionnaires were used in the research, but Cronbach Alpha and other parameters specific to questionnaires were not calculated.

The results in table 3 are not interpreted in such a way as to highlight the differences between the two samples. The present interpretation is only a series of statistical results from which it is not clear what are the differences between the samples for each analyzed parameter.

Conclusions

The conclusions are too general and do not highlight the main results of the study.

The limits of the study and the strengths are usually included in the tests, at the end of them.

Author Response

We would like to thank the reviewers for the quality of their comments, which have been extremely helpful in producing a strengthened version of our manuscript. We also give thanks to the editor for the invitation to deal with all the reviewers’ comments and resubmit an updated version of our manuscript. Please see the attachment: we provide point-by-point responses to the comments carried out by the reviewer 3. In this response, we indicate the modifications carried out in the revised version of the manuscript. The changes have been highlighted in green in order to facilitate the process of review. We hope the manuscript is now acceptable for publication in Behavioral Sciences.

Round 2

Reviewer 1 Report

Well done on addressing my comments and improving the quality of your manuscript.

Reviewer 3 Report

The authors improved the manuscript according with the recommnedations.